# Mother-infant interaction characteristics associate with infant falling reactivity and child peer problems at pre-school age

Silvia Rigato[1]*, Pascal Vrticka[1], Manuela Stets[1], Karla Holmboe[2]

**1** Department of Psychology, Centre for Brain Science, University of Essex, Colchester, England, **2** School of Psychological Science, University of Bristol, Bristol, United Kingdom

* srigato@essex.ac.uk

**Data Availability Statement:** Anonymized individual questionnaire scale scores and interaction coding scores will be publicly available

## Abstract

This longitudinal study investigated the associations between mother-infant interaction characteristics at 9 months of age, maternal mental health, infant temperament in the first year postpartum, and child behaviour at 3 years of age. The infants ($N$ = 54, 22 females) mainly had White British ethnic backgrounds (85.7%). Results showed that i) mother-infant dyadic affective mutuality positively correlated with infant falling reactivity, suggesting that better infant regulatory skills are associated with the dyad's ability to share and understand each other's emotions; and ii) maternal respect for infant autonomy predicted fewer child peer problems at 3 years of age, suggesting that maternal respect for the validity of the infant's individuality promotes better social and emotional development in early childhood.

## Introduction

The quality of caregiver-infant interaction is of fundamental importance for optimal cognitive, social and emotional infant development (e.g., [1–3]), as well as for later behavioural and academic outcomes (e.g., [4,5]). In fact, primary caregivers, mostly mothers in western societies, are key in promoting their infant's development by introducing them to new experiences, supporting emerging skills, and providing opportunities for learning [6]. While mother-infant interactions have been studied extensively in the past decades, it is still unclear which specific components of these interactions positively impact on infant and child development. Similarly, the question of how individual infant characteristics and the wellbeing of the mother interact during their exchanges is still open. In the past decade, the study of mother-infant interactions has increased the use of dyadic measures to capture the nature of such inherently social, bidirectional and reciprocal exchanges more comprehensively. In the present mother-infant interaction study, we employed a coding scheme that comprised of six codes, two relating to maternal behaviour, two to infant behaviour, and two to dyadic interaction behaviour. This approach allowed us to investigate which characteristics of an observed interaction between a mother and her 9-month-old infant were significantly associated with maternal mental health and infant temperament measured throughout the first year postpartum, and which ones predicted child behaviour measured at preschool age.

upon publication. (For peer reviewing purposes, these are already available on https://osf.io/wbpvz/ ?view_only=a4ef97b14830412da791878430673e 0b).

**Funding:** The research was supported by a Grand Challenges Explorations award from the Bill & Melinda Gates Foundation to S.R. and K.H (OPP1119448). https://gcgh.grandchallenges.org/ about The funders had no role in study design, data collection and analysis, decision to publish, or preparation of the manuscript.

**Competing interests:** The authors have declared that no competing interests exist.

Historically, researchers have primarily focused on the mother's sensitive response to infant cues during mother-infant interactions. Ainsworth and colleagues originally defined maternal sensitivity as the ability to perceive and accurately interpret the signals and communications implicit in infant behaviour, and to respond to them appropriately and promptly [7]. Indeed, maternal sensitivity has been found to predict a range of child outcomes, from better emotional regulation [8] to lower levels of aggression [9] and behavioural problems [10]. In addition, attachment theory (e.g., [11]) has traditionally focused on the construct of maternal sensitivity as a main precursor of child secure attachment. However, it is now well-accepted that maternal sensitivity is not the only factor promoting optimal development in young children [12]. The infant's ability to regulate affect (e.g., [13]) as well as mother-child dyadic synchrony during interaction (e.g., [14]) play an important mediating role between maternal sensitivity and infant-mother attachment.

Similarly, another key factor that needs to be considered when observing mother-infant interaction is maternal mental health. Critically, a line of research focusing on the impact of maternal mental health on mother-infant interactions shows how these interactions are affected in clinical as well as sub-clinical samples [15–18]. For example, Cohn et al. [19] reported that depressed mothers have longer latencies in terms of maternal responsivity towards the infant, indicating a lack of synchrony during interactions. In a sample of mothers who were not diagnosed with or treated for depression, Moehler and colleagues (2006) [16] found that depressive symptoms in the first 4 months postpartum were predictive of bonding impairments in the first 14 months of life, with a particularly pronounced impact of maternal depressive symptoms on bonding at 6 weeks postpartum. Another study, conducted in a general population sample with a relatively low number of high-risk mother-infant dyads [20], found an association between depressive symptoms at 6 months postpartum and lower maternal structuring and decreased infant involvement in interaction observed at 8 months (see also [21,22]). The authors concluded that these results may either indicate disruption of interaction patterns that infants of depressed mothers develop early on [23,24] or bidirectional associations between the individual characteristics of the child and maternal caregiving behaviour [25].

Existent literature suggests that maternal and infant characteristics are confounded in parental reports and that these are reflected in the parent-infant relationship [26]. In line with this, Seifer et al. (2004) [27] reported that the mother-infant relationship affects the way the mother perceives her infant's behaviour. Similarly, mothers' expectations and representations of the relationship with their infant may also impact on the way they behave during mother-infant interactions [28]. Indeed, studies have shown that those infants who were perceived as more difficult by their mothers received less responsive mothering during the interaction [29,30]. At the same time, it is also the case that infant characteristics, such as temperament, influence the way an infant takes part in interactions [31]. In a study with 6-month-old infants, parental rating of infant difficult temperament was related to the infant's less responsive behaviour during an interaction with their mother [32]. In addition, a study with mothers and their infants aged between 4 and 11 weeks found that both the mother's intrusiveness and the infant's poor interactive behaviour (i.e., avoidance and lack of active communication and positive vocalisations) significantly increased the infant's likelihood of being perceived as difficult [33]. Another study investigated how maternal and infant characteristics are reflected in the parent-child relationship during the first 3 months of life and showed that maternal unresponsiveness and inflexibility predicted the amount of time an infant fussed and cried at 3 months of age during an observation session [34].

In an interesting investigation by Costa and Figueiredo (2012) [35], infants' behavioural and physiological profiles (i.e., neurobehavioral organization, social withdrawal behaviour, and endocrine reactivity to stress) were analysed in relation to the quality of mother-infant interactions.

The authors concluded that early behavioural characteristics may influence social interaction development, that the mother's behaviour plays a critical role in the modulation of the infant's regulation of biological responses to stressors, and that withdrawn infants may be at risk of developmental difficulties due to both the infants' and mothers' less optimal interaction behaviours.

Taken together, this body of evidence highlights the relevance of studying mother-infant interactions for a better understanding of the dynamic, and likely transactional, associations with maternal mental health and with infant development. Crucially, more recent studies have demonstrated how the behaviours observed in early mother-infant interaction are also related to the child's emotional and behavioural outcomes (e.g., [36,37]) and even to brain development (e.g., [38–40]). For example, in a behavioural study, Mäntymaa et al. (2006) [33] investigated how shared pleasure (defined as the parent and child sharing positive affect in synchrony) in early mother–infant interaction at 2 months was related to later child emotional and behavioural symptoms at 2 years of age. Findings indicated that the duration of shared pleasure was negatively associated with socio-emotional difficulties, i.e., shorter shared pleasure moments were associated with higher infant internalizing and externalizing problem scores at 2 years. Additionally, shared pleasure in early mother–infant interaction was found to be a protective factor in the presence of parental psychopathology. Specifically, children with longer shared pleasure moments in early interaction were more protected against family mental health problems. In children with short shared pleasure moments, internalizing symptoms increased with more psychopathology in the family (both mother's and father's mental health problems), and the risk for higher externalizing problems increased with father's mental health problems. In agreement with Siegel's integrative view of how human development occurs within a social world (2001, 2012) [41,42], Mäntymaa et al. (2006) [33] concluded that children with more optimal dyadic emotion regulation in early infancy are more likely to internalize adaptive emotion regulation models and may be more protected against risks or may show more flexibility in adapting to various demands in the environment. More generally, and in accordance with a transactional perspective, during the transition to toddlerhood, mother-infant interactions lay the foundation for future developmental outcomes (e.g. [43]).

In the current longitudinal study, we aimed to further our understanding of whether and how specific behaviours and dyadic patterns that can be observed when mothers and infants interact are associated with maternal wellbeing and infant characteristics. We specifically focused on associations with maternal depressive symptoms in the first 9 months postpartum, infant temperament in the first 9 months of life, and child behaviour at 3 years of age.

## Methods

### Participants

A total of 81 families were recruited from the University of Essex database and through events for expectant parents at Colchester General Hospital and other local venues in Colchester (UK). The inclusion criteria were that the potential participant was female, pregnant, and had not yet entered the 37th week of pregnancy. There were no exclusion criteria. Ethical approval was received from the National Health Service's National Research Ethics Service Committee, London-Hampstead branch, United Kingdom (15/LO/0478) and from the University of Essex Faculty of Science subcommittee. The approved protocol included a procedure for guiding mothers at high risk of clinical depression to further support and treatment options.

The current study reports on observed mother-infant interaction data (coded from video recordings) and questionnaire data from a longitudinal study investigating attention and social skills in the first year of life, and consisting of one pre- and four post-natal assessment points (36 weeks' gestation, 2 weeks, 4 months, 6 months, 9 months). The prenatal assessment

consisted of a foetal heart rate and movement assessment (data point not included in this article). At each postnatal assessment point, infants took part in behavioural and EEG studies and mothers completed questionnaires relating to their own mental health and to infant temperament. In addition to these measures, at 9 months infants took part in a free-play interaction session with their mothers. A final time point was added at 3 years of age to assess the child's behavioural outcome via questionnaire. While previously published articles include further details on the participant sample and other longitudinal data from these infants [44–46], none includes data from the mother-infant interaction session reported in this article.

Of the 81 recruited families, 16 mothers participated only in the prenatal assessment and 2 infants suffered from complications at birth and were excluded from the final sample. The sample reported on here consisted of 54 infants (22 females) and their mothers who came to the lab to take part in the mother-infant interaction session when the infant was 9 months old. No formal sample size calculation was used due to the lack of previous longitudinal studies with similar measures to base the calculation on. There were also financial and time constraints. However, a sample size of 54 infants is larger than most infant studies involving extensive behavioural testing. The majority of participating infants were born full-term (at 37 weeks of gestation or later; $M$ = 40 weeks, 3 days), and only two infants were born between 36 and 37 weeks of gestation. Infants in the sample had normal birth weight ($M(SD)$ = 3.5(0.4) Kg; Range = 2.44–4.82 Kg), no complications at birth, and no known health issues (pregnancy, birth and health information was missing for 4 infants). At the time of the prenatal assessment, which occurred around 36 weeks of gestation, mothers were on average 31.2 years of age ($SD$ = 4.6) and fathers were on average 33.6 years of age ($SD$ = 6.1). Mothers had spent an average of 16.8 years in education ($SD$ = 3.6) and fathers had spent an average of 15.2 years in education ($SD$ = 3.7). Participating families had an average income of £54,483 ($SD$ = £28,606). The infants' ethnic background was mainly White British ($N$ = 54; 85.7%). Three infants were of Other White background (4.8%), four infants had mixed ethnic background but did not provide further details (6.4%), one infant was White British mixed with another ethnic background (1.6%), and one infant had a Hispanic background (1.6%).

## Questionnaires

Participating mothers were asked to complete well-established questionnaires relating to infant temperament as well as symptoms of depression. Details on each of the questionnaires are provided below. The links to the online questionnaires were sent via email to those families who had confirmed attendance for a scheduled lab test session for the following day in an effort to keep the time difference between test session and questionnaire completion as short as practically possible.

**The Beck Depression Inventory, 2nd Edition.** To assess mothers' level of depressive symptoms over the course of their infant's first year, we asked them to complete the Beck Depression Inventory, 2nd Edition (BDI-II; [47]). The BDI-II is a standardized questionnaire containing 21 items. Each question asks to what extent a certain state does or does not apply to the respondent (e.g., Sadness; response options: 0 –I do not feel sad, 1 –I feel sad much of the time, 2 –I am sad all the time, 3 –I am so sad or unhappy that I can't stand it). A higher score indicates a higher level of depressive symptoms. A review of 118 studies established high internal consistency of the BDI-II, with Cronbach's alpha ranging from .83 to .96 across both clinical and non-clinical samples [48]. In the current study, Cronbach's alpha for BDI-II scores at the four assessment time points ranged from .87 to .94 (see S1 Table in S1 File), with a mean alpha of .92. BDI-II questionnaires were scored by adding up the ratings for all items with the first option always scoring zero and the last option always scoring three. The sum of scores indicates four different levels of

depression. A score between 0 and 13 is classified as "minimal depression". A score between 14 and 19 is classified as "mild depression". A score between 20 and 28 is classified as "moderate depression", and a score between 29 and 63 is classified as "severe depression" [47].

**The Infant Behavior Questionnaire–Revised – Very Short Form.** Mothers completed the Infant Behavior Questionnaire–Revised–Very Short Form (IBQ-R VSF; [49]), as well as two additional scales from the Infant Behavior Questionnaire–Revised (IBQ-R; [50]). The IBQ-R VSF consists of 37 items in total, and can be separated into three broad scales: Surgency (13 items; e.g., "When tossed around playfully, how often did the baby laugh?"), Negative Affect (12 items; e.g., "At the end of an exciting day, how often did your baby become tearful?"), and Orienting/Regulatory Capacity (12 items; e.g., "How often during the last week did the baby enjoy being read to?") [49]. In addition to these scales, we also included the scales for Distress (12 items; e.g., "After sleeping, how often did the baby fuss or cry immediately?") and Falling Reactivity (13 items; e.g., "When put down for a nap, how often did your baby settle down quickly?") from the IBQ-R [50]. For each question, mothers were asked to rate how often their infant had shown a specific behaviour in a given situation during the past week by selecting a score between 1 ("Never") and 7 ("Always"). Additionally, an item could be scored as "NA–Does not apply" should the situation described in the question not have occurred over the previous seven days. Average scores were calculated for each scale. Items rated as "NA–Does not apply" were excluded from analysis.

Putnam et al. (2014) [49] reported an average Cronbach's alpha of .75 or above for the three broad IBQ-R VSF scales. In the current study, Cronbach's alpha for the Surgency scale ranged from .70 to .91 across ages, with a mean alpha of .80; Cronbach's alpha for the Negative Affect scale ranged from .79 to .88 across ages, with a mean alpha of .82; Cronbach's alpha for the Orienting/Regulatory Capacity scale ranged from .62 to .80 across ages, with a mean alpha of .71. Montirosso et al. (2011) [51] reported average Cronbach's alphas of .83 and .84 across two samples for the variables of Distress and Falling Reactivity, respectively. An overview of a number of studies using different versions of the IBQ-R revealed that Cronbach's alpha for Distress and Falling Reactivity ranged between .74 and .79 ($M = .76$) and between .76 and .84 ($M = .80$), respectively, when using the short version of the IBQ-R [47]. In the current study, Cronbach's alpha for the Distress scale ranged from .59 to .70 across ages, with a mean alpha of .64, and Falling Reactivity ranged from .63 to .85 across ages, with a mean alpha of .78 (see S2 Table in S1 File).

**The Strengths and Difficulties Questionnaire (SDQ).** To investigate the child's behaviours, emotions and relationships, mothers were asked to complete the Strengths and Difficulties Questionnaire (SDQ, [52]) when the child was 36 months old. This consists of 25 items separated into 5 subscales. In the current study, Cronbach's alpha for the five SDQ subscales were as follows: .65 for Emotional Symptoms, .43 for Conduct Problems, .71 for Hyperactivity-Inattention, .52 for Peer Problems, and .74 for Prosocial Behavior. It should be noted that Cronbach's alpha was acceptable for all subscales in the large validation sample of 5- to 15-year-olds in Goodman (2001) [53]. The lower alpha for some SDQ scales in our sample could be due to younger children showing less differentiation of behavioural problems. Furthermore, given that all the subscales, except the Hyperactivity-Inattention scale, were not normally distributed (please refer to Supplementary Material in [45]), the lower alphas could also reflect low variance of symptoms in the population sampled, although Goodman et al. (2001) [53] also assessed a general population sample.

## Lab observation of mother-infant interaction

Mothers were invited to the lab when their infants were 9 months old. They were shown a play area with age-appropriate toys and books on a floor mat, and asked to play with their infants for a short while (about 10 minutes) as they would normally do at home. All experimenters left

the room during this playtime, and the dyads were video recorded with two cameras focusing on the mother and infant, respectively.

**Coding of interaction behaviours.** Six consecutive minutes of interaction were coded in one-minute intervals on a 7-point Likert scale. The coding was started from the moment the experimenter was heard to have left the room. The coding system used for this study was adapted from scales previously used [54–58]. It comprised six codes, two relating to maternal behaviour, two to infant behaviour, and two to dyadic interaction behaviour. The maternal rating scales included maternal supportive presence (MSP) and maternal respect for infant autonomy (MRIA). The MSP scale concerns the mother's level of emotional support towards the infant; a high score on this scale would be characterised by expressions of encouragement or positive regard as well as through physical proximity. The MRIA scale reflects the degree to which the parent recognizes and respects the validity of the infant's individuality, motives, and perspectives. For example, a parent scoring high on this scale would be verbalizing her acknowledgement of the infant's intentions and negotiating the course of the interactions. The infant rating scales included infant agency / autonomy (IA)–the degree to which the infant shows agency and enthusiasm in the interaction, takes an active interest in his or her activities, invests effort in them, and appreciates successes–and infant negativity (IN)–the amount of negative affect and emotions toward the mother displayed during the interaction. Infants scoring high in IA would show confidence and eagerness to play, while infants high in IN would show anger, dislike, or hostility toward the parent. The dyadic interaction scales included dyadic affective mutuality (DAM), i.e. how strongly the mother-infant dyad shared and understood each other's emotions, and dyadic reciprocity (DREC), i.e. how strongly the mother's and infant's behaviour and verbalisations during the interaction were reciprocated in a conversation-like manner. Specifically, DAM assesses the availability and mutuality of emotion between infant and parent as well as how secure the infant feels with the parent. Dyads high on this scale almost always have a moment of shared pleasure. DREC is characterised by contingent responsiveness, i.e., well-timed behavioural or verbal responses to comments, questions, suggestions or non-verbal cues from the parent and/or the infant. A detailed description of the coding scheme specifically adapted for this study, with examples for each code, can be found in S1 File.

Six undergraduate students, unaware of the questionnaire data, were trained in coding using 10% of the mother-infant interaction videos. They subsequently coded the remaining 90% of videos, with an additional 10% of these videos coded by at least two students to enable continuous interrater reliability monitoring. The final interrater reliability was then calculated for the complete set of videos (i.e., 20%) that were double coded, and yielded the following intraclass correlation (ICC) values: MSP = .581, MRIA = .681, IA = .724, IN = .748, DAM = .741, DREC = .764. Accordingly, apart from the MSP code, all codes either reached acceptable or good interrater reliability. We used all codes for subsequent analyses, but acknowledge that results involving maternal codes, and particularly MSP, should be regarded with some caution due to their relatively lower interrater reliability.

## Results

Maternal depression symptom scores obtained from the BDI-II [47], infant temperament scale scores obtained from the IBQ-R VSF [49] and IBQ-R [50], the child behavioural outcome scores obtained from the SDQ [52], and the interaction behaviours obtained from the interaction coding scales are summarised in Table 1 below. All relevant variables were checked for distribution and outliers. Many variables did not meet normality criteria due to (left or right) skew. For subsequent simple correlation analyses, we therefore used Spearman's *r*, and multiple regression analyses were always checked for normality (e.g., residual distributions). As

several variables included extreme outliers (values beyond 3 standard deviations from the mean), these outliers were winsorised up or down to the respective upper or lower 3 standard deviation boundary before analysis. Each analysis was initially run once with and once without including sociodemographic variables (family annual income, maternal age, maternal years in education, and child gestational age). Since both analyses yielded comparable results, here we only report the analyses without the sociodemographic variables. The analyses including the sociodemographic variables can be found in S8-S11 Tables in S1 File.

## Associations between interaction behaviours

Simple correlations, Bonferroni-corrected for multiple comparisons (adjusted p-value q< .003), were run to investigate the associations between the interaction coding scales of maternal behaviour (MSP and MRIA), infant behaviour (IA and IN), and dyadic behaviour (DAM and DREC). Results revealed a significant positive correlation between the two maternal behaviour scales MSP and MRIA ($r = .518$, $p < .001$), and between the two dyadic behaviour scales DAM and DREC ($r = .568$, $p < .001$). Furthermore, the infant behaviour scale IA correlated positively with the two dyadic behaviour scales DAM ($r = .417$, $p = .002$) and DREC ($r = .454$, $p < .001$). Results are summarised in S3 Table in S1 File.

## Associations between interaction behaviours and maternal depressive symptoms

Correlational analyses were conducted to test for associations between maternal depressive symptom scores derived from the BDI-II [47], averaged across the first 9 months, and

**Table 1. Average (SD) scores of the measures used in this study.**

| | | |
|---|---|---|
| *Maternal depressive scores (BDI-II)* *Average (SD) 0–9 months postpartum* | 8.4 (6.7) | |
| *Infant temperament scales (IBQ-R VSF, IBQ-R)* *Average (SD) 0–9 months* | *Negative Affect* | 3.8 (0.8) |
| | *Distress* | 3.6 (0.6) |
| | *Falling Reactivity* | 4.9 (0.8) |
| | *Surgency* | 3.9 (0.6) |
| | *Orienting / Regulatory Capacity* | 5.1 (0.6) |
| *Child Strengths and Difficulties Questionnaire* *Average (SD) at 3 years* | *Emotional Symptoms* | 1.2 (1.5) |
| | *Conduct Problems* | 2.5 (1.1) |
| | *Hyperactivity-Inattention* | 3.5 (1.7) |
| | *Peer Problems* | 3 (1.8) |
| | *Prosocial Behavior* | 7.2 (1.8) |
| *Interaction coding scales* *Average (SD) at 9 months* | *Maternal Supportive Presence* | 4.2 (0.8) |
| | *Maternal Respect for Infant Autonomy* | 4.6 (0.8) |
| | *Infant Agency / Autonomy* | 2.6 (1.0) |
| | *Infant Negativity* | 1.3 (0.4) |
| | *Dyadic Affective Mutuality* | 3.1 (0.8) |
| | *Dyadic Reciprocity* | 2.4 (0.7) |

*Note*. Maternal depressive symptom scores obtained from the Beck Depression Inventory, 2nd Edition (BDI-II, [47]), infant temperament scale scores obtained from the Infant Behavior Questionnaire–Revised–Very Short Form (IBQ-R VSF; [49]) and from the Infant Behavior Questionnaire–Revised (IBQ-R; [50]), the child behavioural outcome scores obtained from the Strengths and Difficulties Questionnaire (SDQ, [52]), and the interaction behaviours obtained from the interaction coding scales.

behaviours obtained from the interaction coding scales. Because there was only one independent variable in this analysis (i.e., BDI-II scores), we used a simple Spearman correlation. No significant associations were found. Results are summarised in S4 Table in S1 File. However, when socio-economic status variables (family annual income, maternal age, maternal years in education, and child gestational age) were included in the analyses, we observed a negative association between maternal depressive symptoms and MRIA (r = -.297, p = .047) and a positive association between maternal depressive symptoms and IN (r = .335, p = .024). Neither of these correlations, however, remained significant after multiple comparisons correction. Results are summarised in S9 Table in S1 File.

### Associations between interaction behaviours and infant temperament

Regression analyses were conducted to explore associations between the 6 interaction behaviour coding scales and the 5 infant temperament scales from the IBQ-R VSF [49] and IBQ-R [50]. The scores of the temperament scales were averaged across the 4 assessment time points in the first 9 months of life. We conducted one multiple regression analysis for each interaction coding scale and subsequently Bonferroni-corrected all significance values for multiple comparisons–i.e., the number of interaction coding scales (adjusted p-value q < .0083). The overall multiple regression model was significant for the two dyadic interaction codes DAM, p < .007, and DREC, $p < .001$, but not for any of the maternal or infant codes. DAM was positively associated with Falling Reactivity (p = .004), and no other significant association was found with any of the other temperament scales. DREC was associated with Orienting / Regulatory Capacity (p = .04) and with Falling Reactivity (p = .05) but after controlling for multiple comparisons, these associations were no longer significant. Results of the significant multiple regression models are summarised in Table 2, and the significant association between DAM and Falling Reactivity is illustrated in Fig 1. Results of all the multiple regression models are summarised in S5 Table in S1 File.

An additional correlation analysis, conducted to better understand at what time point(s) in the first months of life infant Falling Reactivity was associated with DAM (observed during the interaction at 9 months) showed that all correlations were significant except the one involving Falling Reactivity at 2 weeks (see S6 Table in S1 File).

### Associations between interaction behaviours and child behavioural outcomes

Regression analyses were conducted to explore associations between the 6 interaction behaviour coding scales and the 5 child behavioural outcomes as assessed with the SDQ [52] at 3 years of age. We conducted one multiple regression analysis for each interaction coding scale. Only the overall multiple regression model for MRIA was significant (p = .03) (Table 3), although it did not survive correction for multiple comparisons (adjusted p-value q < .0083). This revealed a significant association between MRIA at 9 months and Peer Problems at 3 years (p = .003) (Fig 2). None of the other multiple regression models was significant (See S7 Table in S1 File).

### Discussion

Existent literature suggests that the behaviours observed in mother-infant interactions are associated with both maternal mental health and infant individual characteristics, and they also predict later socio-emotional child outcomes. However, it is unclear which components and dyadic patterns of the observed behaviours associate with each of these characteristics and outcomes. In this longitudinal study, we observed mothers interacting with their 9-month-old

**Table 2. Associations between dyadic interaction behaviours and infant temperament.**

*Model Fit Measures for Dyadic Affective Mutuality (DAM)*

| Model | R | R$^2$ | Adjusted R$^2$ | Overall Model Test | | | |
| --- | --- | --- | --- | --- | --- | --- | --- |
| | | | | F | df1 | df2 | p |
| 1 | 0.525 | 0.275 | 0.200 | 3.65 | 5 | 48 | **0.007** |

*Multiple Regression Coefficients for Dyadic Affective Mutuality (DAM)*

| Predictor | Estimate | SE | t | p |
| --- | --- | --- | --- | --- |
| Intercept | 1.9498 | 1.432 | 1.362 | 0.180 |
| Negative Affect | -0.2801 | 0.181 | -1.547 | 0.128 |
| Surgency | 0.1137 | 0.204 | 0.558 | 0.580 |
| Orienting / Regulatory Capacity | 0.0940 | 0.208 | 0.452 | 0.653 |
| Distress | 0.1792 | 0.258 | 0.695 | 0.490 |
| Falling Reactivity | 0.5067 | 0.166 | 3.057 | **0.004** |

*Model Fit Measures for Dyadic Reciprocity (DREC)*

| Model | R | R$^2$ | Adjusted R$^2$ | Overall Model Test | | | |
| --- | --- | --- | --- | --- | --- | --- | --- |
| | | | | F | df1 | df2 | p |
| 1 | 0.594 | 0.352 | 0.285 | 5.22 | 5 | 48 | **< .001** |

*Multiple Regression Coefficients for Dyadic Reciprocity (DREC)*

| Predictor | Estimate | SE | t | p |
| --- | --- | --- | --- | --- |
| Intercept | -0.9950 | 1.196 | -0.832 | 0.409 |
| Negative Affect | 0.1721 | 0.151 | 1.138 | 0.261 |
| Surgency | 0.0527 | 0.170 | 0.309 | 0.758 |
| Orienting / Regulatory Capacity | 0.3617 | 0.174 | 2.083 | 0.043 |
| Distress | -0.1755 | 0.215 | -0.815 | 0.419 |
| Falling Reactivity | 0.2750 | 0.138 | 1.986 | 0.053 |

*Note*. Dyadic Affective Mutuality (DAM) and Dyadic Reciprocity (DREC) obtained from the interaction coding scales and infant temperament scales obtained from the Infant Behaviour Questionnaire–Revised–Very Short Form (IBQ-R VSF; [49]) and from the Infant Behaviour Questionnaire–Revised (IBQ-R; [50]). Significant predictors are in bold.

infants and employed an interaction coding scheme that comprised codes specific to behaviours of each interaction partner as well as to the dyadic interaction behaviour. This allowed us to investigate how each of these behaviours associated with i) maternal depressive symptoms in the first 9 months postpartum, ii) infant temperament in the first 9 months of life and iii) child behaviour at 3 years of age. Our results indicated 1) a lack of association between interaction behaviours and maternal depressive symptoms, 2) a positive association between infant falling reactivity and dyadic affective mutuality, and 3) a negative association between maternal respect for infant autonomy and child peer problems at 3 years of age. We will discuss each of these findings below.

Our analyses focusing on maternal depressive symptoms did not reveal any significant association with the behaviours observed during the interaction session when socio-economic status (SES) variables (family annual income, maternal age, maternal years in education, and child gestational age) were excluded from the analyses. This is interesting as previous literature shows that mothers' mental health difficulties may profoundly affect their ability to engage in dyadic interactions with their infants. For example, mothers suffering from depressive symptoms might show less sensitivity or responsiveness, and more difficulties in reading their infant's facial expressions, affecting their ability to attune to their infant's internal states [59–61]. Previous studies have also found that dyads where the mother is depressed spend more

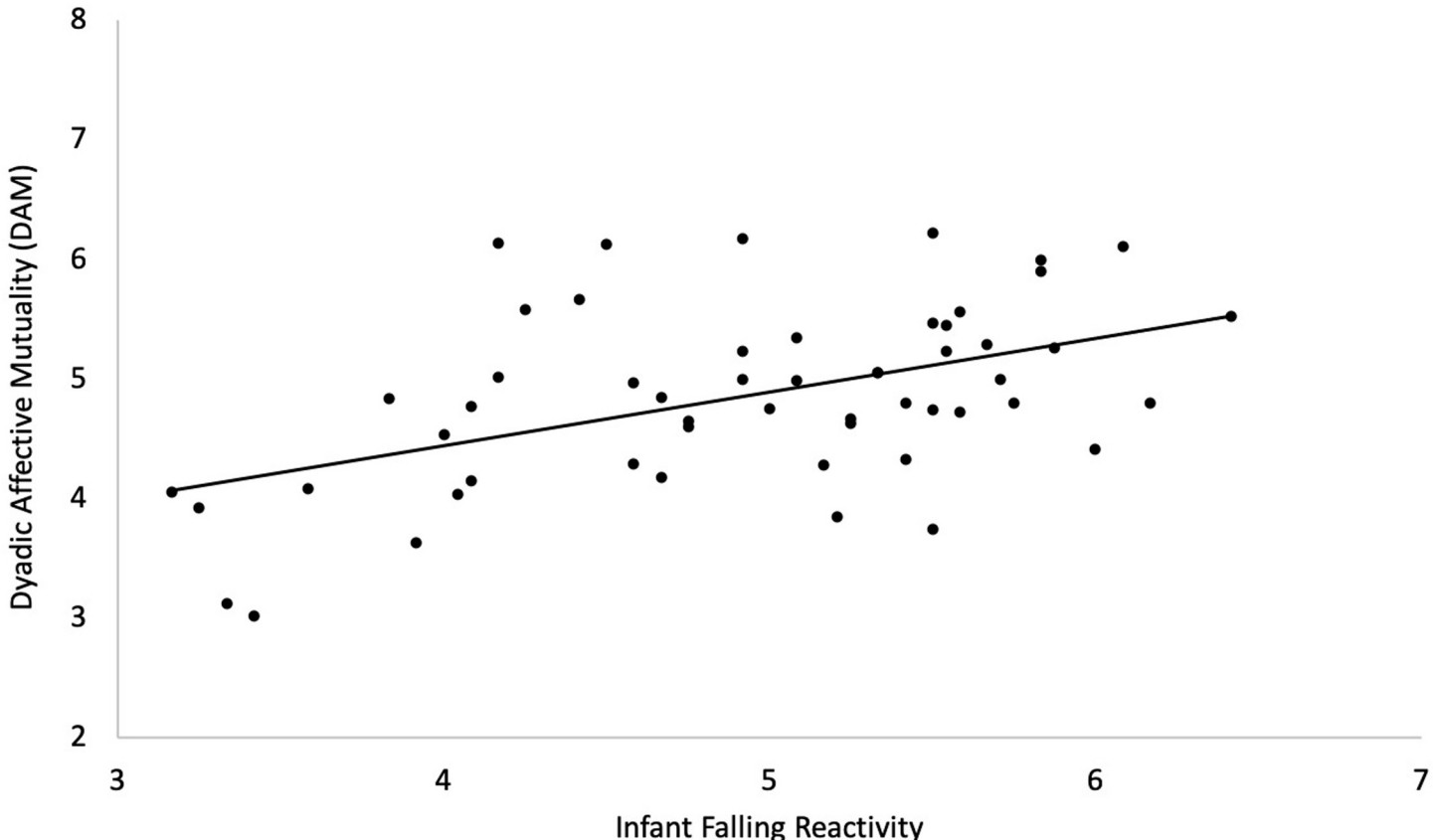

**Fig 1. Illustration of the significant association between Dyadic Affective Mutuality (DAM) and infant falling reactivity.** *Note.* Dyadic Affective Mutuality (DAM) obtained from the interaction coding scales and infant Falling Reactivity measured with the Infant Behavior Questionnaire–Revised (IBQ-R; [50]).

time in negative behavioural states and that the mother tends to react more negatively to their infant's negative behaviour [23,62]. Our own previous findings from this sample, based on

**Table 3. Associations between Maternal Respect for Infant Autonomy (MRIA) and child behavioural outcomes.**

*Model Fit Measures for Maternal Respect of Infant Autonomy (MRIA)*

| | | | | Overall Model Test | | | |
|---|---|---|---|---|---|---|---|
| **Model** | **R** | **R²** | **Adjusted R²** | **F** | **df1** | **df2** | **p** |
| 1 | 0.515 | 0.265 | 0.168 | 2.74 | 5 | 38 | 0.033 |

*Multiple Regression Coefficients for Maternal Respect of Infant Autonomy (MRIA)*

| **Predictor** | **Estimate** | **SE** | **t** | **p** |
|---|---|---|---|---|
| Intercept | 5.5519 | 0.5973 | 9.367 | < .001 |
| Emotional Symptoms | -0.0082 | 0.0906 | 0.188 | 0.929 |
| Conduct Problems | -0.1093 | 0.1201 | -1.012 | 0.368 |
| Hyperactivity-Inattention | 0.1140 | 0.0756 | 1.577 | 0.140 |
| Peer Problems | -0.2189 | 0.0685 | -3.291 | **0.003** |
| Prosocial Behaviour | -0.0645 | 0.0633 | -1.018 | 0.315 |

*Note.* Maternal Respect for Infant Autonomy (MRIA) obtained from the interaction coding scales and child behavioural outcome obtained from the Strengths and Difficulties Questionnaire (SDQ, [52]). Significant predictors are in bold.

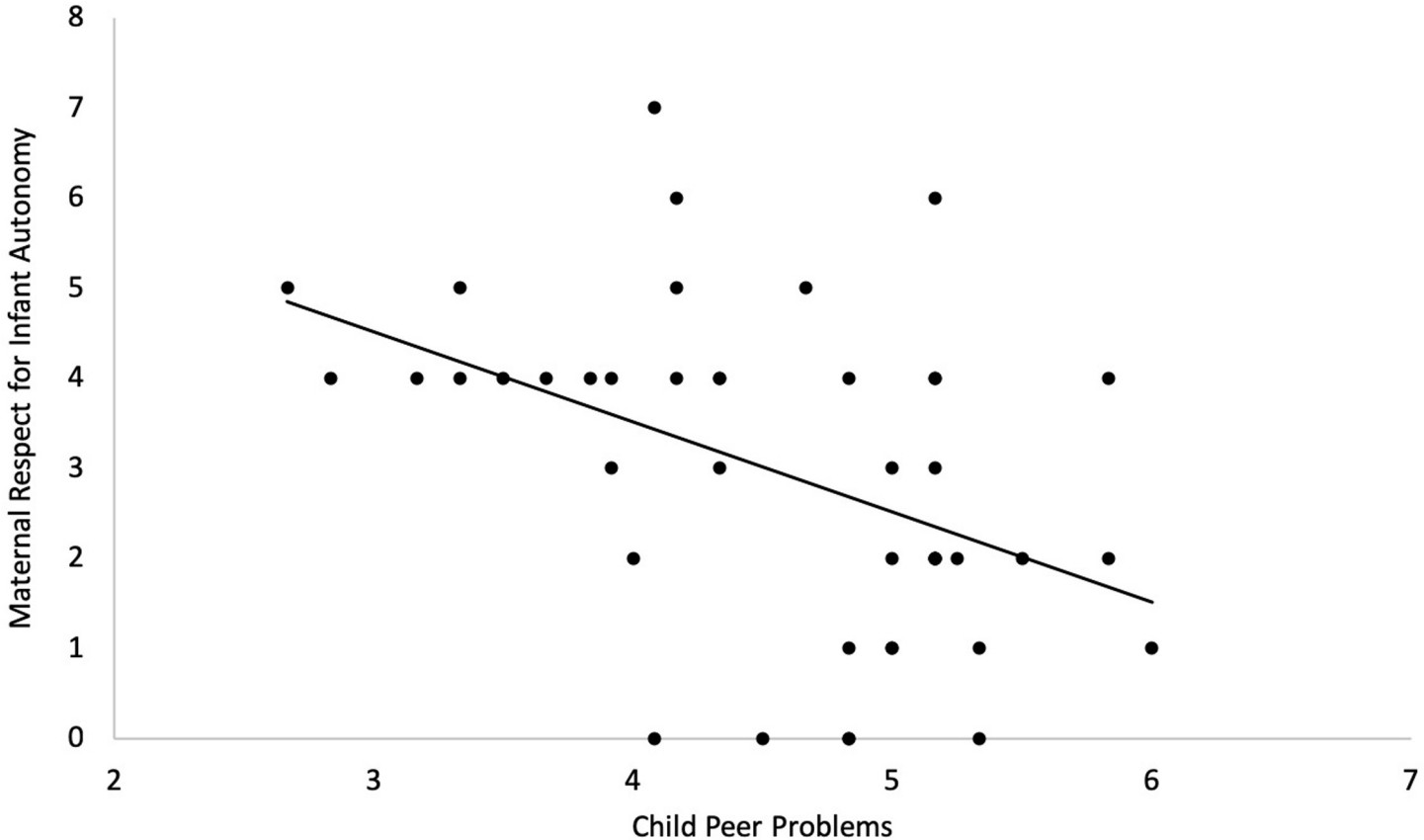

**Fig 2. Illustration of the significant association between Maternal Respect for Infant Autonomy (MRIA) and child peer problems.** *Note*. Maternal Respect for Infant Autonomy (MRIA) obtained from the interaction coding scales and Peer Problems as measured by the Strengths and Difficulties Questionnaire (SDQ, [52]).

questionnaire data alone [44], showed cascading associations between maternal depressive symptoms and infant negative affect across the first year of life, and primarily during the first 6 months of life, with most of these effects being maternally driven (i.e., earlier maternal depression predicted later infant negative affect, but not the other way round). Other previous studies have also found that infants of depressed mothers are characterized by more negative affect [e.g., 60,63,64], and a recent study replicated the association between maternal depression and infant negative affect at 10 months, whereas this association had disappeared by 16 months [65]. Thus, this specific association between maternal depressive symptoms and the construct of infant negative affect might be stronger in early infancy and start weakening by the end of the first year of life. It is possible that the coding scales employed in this study were not focused on the same elements that defined maternal sensitivity or responsiveness in previous studies, nor infant negative affect as measured in the IBQ-R VSF [49]. For example, infant negativity in the interaction coding scheme employed here was assessed as the degree to which the infant showed anger, dislike, or hostility toward the parent. This is indeed different from what is measured in the IBQ-R VSF [49] where infant negative affect relates to infants' reactions in unfamiliar or potentially stressful situations. Another key difference is that the IBQ-R VSF [49] measure is based on observations across multiple different situations summarised by a single score. The mother-infant interaction observed in this study, on the other hand, provides a snapshot of behaviours in a very specific situation and may therefore not tap into the full range of negative affect behaviours that would be observed across a range of circumstances.

Despite these differences, it is worth noting that when demographics of SES (family annual income, maternal age, maternal years in education, and child gestational age) were considered in the analyses, we found a positive association between maternal depressive symptoms reported in the first 9 months post-partum and infant negativity during the interaction observed in this study. We also found a negative association between maternal depressive symptoms and maternal respect for infant autonomy. While neither of these correlations remained significant after multiple comparisons correction, existent literature suggests that SES is an important factor that impacts on both maternal mental health, generally reporting higher SES levels associated with better maternal mental health (e.g. [66]), and on child development [67,68]. For example, high parental educational level and family income are key protective components of the parent–child relationship [69], and are also related to developmental outcomes, such as language and literacy skills [70], fine motor development [71] and school readiness [72]. SES can impact child development also indirectly by influencing proximal factors, such as parenting attitudes. For example, research suggests that higher maternal education level has a positive influence on parenting strategies because children are more likely to be exposed to complex vocabulary and books, and mothers are more likely to have greater knowledge of child development and higher educational expectations for her own children [73–75]. Our findings suggest that when SES variables, such as family annual income, maternal age, maternal education, and child gestational age, are included in the correlational analyses, higher maternal depressive symptoms are associated with higher infant negative affectivity and also with fewer occurrences of maternal respect of the validity of the infant's individuality, desires and perspectives during the interaction.

Our second set of analyses focused on associations between observed behaviours in the interaction and infant temperament. The results suggest that the dyad's ability to share and understand each other's emotions is associated with higher infant falling reactivity in the first year of life. Falling reactivity describes to what extent an infant is capable of recovering from a high level of distress or even positive excitement and, therefore, the infant's ability to self-regulate [50]. Feldman (2003, 2007) [76,77] emphasized the importance of sharing positive affect in parent–infant interaction, and claimed that synchronized positive affect is essential for the development of self-regulation. The dyad's ability to share and communicate emotions to each other, alongside the caregiver's empathic understanding of the infant's affective state, are indeed recognised steps in the process of emotion regulation in the literature [e.g., 78,79]. While it is difficult to define a sole directionality of the association between infant temperament and the dyad's behaviour, our data hint to an infant-driven effect where infants with high self-regulation skills from an early age engage with the parent in a more positive way. Indeed, our additional correlational analyses (reported in S1 File) revealed that infant falling reactivity predicts later dyadic affective mutuality (here observed at 9 months) from 4 months of age. Critically though, in order to establish the directionality of these associations, future studies should include mother-infant observations also at earlier time points. The dyadic affective mutuality code employed in this study specifically assessed availability and mutuality of emotion between mother and infant, how secure the child felt with the parent, and verbal and non-verbal communication of the two interaction partners. In dyads high on this scale, both emotion and communication flew freely, and smiles, eye contact, proximity seeking and help seeking behaviours were frequently observed. There was at least one shared occurrence of reciprocally communicated positive emotion, however, both positive and negative emotions were shared. With this study, we show that infants who have better abilities to cope with distress and/or overwhelming emotions, and are therefore better able to self-regulate, experience more of this kind of positive interaction with their caregiver.

Finally, our findings suggest that maternal respect for infant autonomy promotes a better social and emotional development in early childhood. Specifically, we found that infants whose mothers recognized and respected the validity of the infant's individuality, desires and perspectives during the interaction, go on to develop fewer peer problems at preschool age. One interpretation of this finding is that infants learn through the interactions with their mothers how to respect other people's autonomy and identity, which in turn may help them make friends, be liked by others and negotiate challenging social situations. Autonomy is considered a fundamental child need and parental support for autonomy is arguably one of the most important aspects of parenting for the development of child independence [80]. Requests for autonomy are often found among the most frequent cues in toddlers' interactions with both their mother and father (e.g. [81]), highlighting the emerging need for autonomy in the transition from infancy to toddlerhood. Similar to Mäntymaa et al.'s (2006) [33] finding that shared pleasure in early mother–infant interaction was a protective factor against internalizing and externalizing problems at 2 years, our results suggest that maternal respect for infant autonomy in the first year of life acts as a protective factor against child peer problems at 3 years of age. A study with older children also reported that a longer duration of positive interactions between parent and child at 3 years was associated with fewer conduct problems at 4 years [82], therefore supporting the view that more positive early interactions are associated with lower levels of socio-emotional and behavioural problems later in development. These conclusions are also supported by a meta-analysis by Pinquart (2017) [83] who examined the relations between parent-child interactions and child externalizing behaviours. This work reports unidirectional longitudinal associations between increases in parental support and lower levels of externalizing behaviours [83]. Interestingly, Hughes et al. [84] also report an inverse association between maternal autonomy support at 14 months and externalizing behaviour at 2 years but only for those infants who showed low levels of negative affect at 4 months. The authors discuss their findings in relation to models of vantage sensitivity [e.g., 85], i.e., infants with easy temperaments may gain most from high quality maternal support. However, they also suggest the possibility that their findings could reflect child-driven effects [86], i.e., mothers of infants prone to distress may have attempted to minimize frustration by providing greater assistance during the interaction and thereby inadvertently limited their child's autonomy. In addition, another recent study [87] reported differential patterns of associations between toddler temperament, parental stress and support for autonomy depending on the temperamental characteristics. Parent stress was found to fully account for the negative association between toddler negative affectivity and parent autonomy support, suggesting that toddler temperament affects parent wellbeing, which in turn is related to parenting practices. While our sample size does not allow for testing these interpretations, taken together the evidence shows that parental autonomy support is crucial for healthy child adjustment [88], and in particular during toddlerhood [81,87].

Our work is not without limitations. For instance, a larger sample size would have allowed for deeper investigation of the complex inter-relations among the interaction elements and the individual temperamental characteristics of infants and their mothers' wellbeing. Collecting information from additional observers of the child (e.g., other family members, educators, etc.) would have allowed a more objective assessment of the infant temperament as well as the child's social and emotional outcome at 3 years of age. Further, our study only included a single observation of mother-infant interaction, therefore limiting the generalizability of these findings to a specific point in time. The dynamics in play during mother-infant interactions are likely to change with infant age, making it important to track how these changes are related to more stable temperamental traits and maternal mental health. Future research should therefore include a larger sample and a mother-infant interaction protocol at multiple ages. This

would also allow for investigation of potential bi-directional and transactional effects over time. Relatedly, collecting measures of prenatal maternal depressive symptoms as well as considering other (prenatal) biological factors will also be important in future studies. Finally, while the focus of this study was on the interactions infants experience with their mother, it would also be valuable to further our understanding of how interactions with fathers and other caregivers influence infant and child development.

Despite these limitations, with this study we show important links between infant falling reactivity and the exchanges of emotion between mother and infant, and between maternal respect for infant autonomy and child social adjustment at 3 years of age. This type of information is important for defining and implementing intervention strategies that will facilitate and promote a healthy mother-infant relationship from the very beginning. For example, given the association between the mother's respect for infant autonomy and her child's later social development, interventions could aim to enhance mothers' attention to the acknowledgement of her infant's intentions and desires. Similarly, mothers could benefit from being aware of the importance of sharing their affective states with their infants and of the association with infant self-regulatory abilities. Altogether, a better understanding of the transactional associations between maternal wellbeing, infant temperament, and interaction behaviours could benefit the development of healthier mutual relationships.

## Supporting information

**S1 File.**
(DOCX)

## Acknowledgments

We would like to thank all the families who took part in the study and thereby made this research possible.

## Author Contributions

**Conceptualization:** Silvia Rigato, Karla Holmboe.

**Data curation:** Silvia Rigato, Pascal Vrticka, Manuela Stets, Karla Holmboe.

**Formal analysis:** Silvia Rigato, Pascal Vrticka.

**Funding acquisition:** Silvia Rigato, Karla Holmboe.

**Investigation:** Silvia Rigato, Manuela Stets, Karla Holmboe.

**Methodology:** Silvia Rigato, Karla Holmboe.

**Project administration:** Manuela Stets.

**Supervision:** Silvia Rigato, Pascal Vrticka, Manuela Stets, Karla Holmboe.

**Writing – original draft:** Silvia Rigato.

**Writing – review & editing:** Silvia Rigato, Pascal Vrticka, Manuela Stets, Karla Holmboe.

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
