## [Decision Letter · Decision Letter 0]

29 Jan 2024

PONE-D-23-29119Mother-infant interaction characteristics associate with infant falling reactivity and child peer problems at pre-school agePLOS ONE

Dear Dr. Rigato,

Thank you for submitting your manuscript to PLOS ONE. After careful consideration, we feel that it has merit but does not fully meet PLOS ONE’s publication criteria as it currently stands. Therefore, we invite you to submit a revised version of the manuscript that addresses the points raised during the review process.

We look forward to receiving your revised manuscript.

Kind regards,

Ammal Mokhtar Metwally, Ph.D (MD)

Academic Editor

PLOS ONE

Journal Requirements:

Additional Editor Comments:

The manuscript is interested meanwhile, the reviewers have raised a number of points which we believe would improve the manuscript and may allow a revised version to be published in PLOS one.

Reviewers' comments:

Reviewer's Responses to Questions

**Comments to the Author**

1. Is the manuscript technically sound, and do the data support the conclusions?

Reviewer #1: Yes

Reviewer #2: Yes

2. Has the statistical analysis been performed appropriately and rigorously? 

Reviewer #1: Yes

Reviewer #2: Yes

3. Have the authors made all data underlying the findings in their manuscript fully available?

Reviewer #1: Yes

Reviewer #2: Yes

4. Is the manuscript presented in an intelligible fashion and written in standard English?

Reviewer #1: Yes

Reviewer #2: Yes

5. Review Comments to the Author

Reviewer #1: Thank you for the opportunity to review the manuscript, “Mother-infant interaction characteristics associate with infant falling reactivity and child peer problems at pre-school age”. PONE-D-23-29119

The topic of the manuscript is interesting, the quality of the early interaction between an infant and his or her mother is an important factor that influences the child’s development, survival and growth. This type of research has been discussed repeatedly in previous literature; however, it is still needed to direct the plans for intervention procedures to alter mother–infant interaction with the ultimate goal of achieving positive developmental outcomes during the first years of life.

The introduction

The introduction must be revised as there are some outdated references. It is prolonged and full of details of previous studies. It included description of the tools which have been used in the study. These details must be transferred to the methodology section.

These are examples of recent references that can help in revising the introduction

• Rocha, N. A. C. F., Dos Santos Silva, F. P., Dos Santos, M. M., & Dusing, S. C. (2020). Impact of mother-infant interaction on development during the first year of life: A systematic review. Journal of child health care: for professionals working with children in the hospital and community, 24(3), 365–385. https://doi.org/10.1177/1367493519864742

• Chung, FF., Wan, GH., Kuo, SC. et al. Mother–infant interaction quality and sense of parenting competence at six months postpartum for first-time mothers in Taiwan: a multiple time series design. BMC Pregnancy Childbirth 18, 365 (2018). https://doi.org/10.1186/s12884-018-1979-7

• Richter M, Fehringer K, Smith J, Pineda R. Parent-infant interaction in the NICU: Challenges in measurement. Early Hum Dev. 2022 Jul;170:105609. doi: 10.1016/j.earlhumdev.2022.105609. Epub 2022 Jun 14. PMID: 35752043; PMCID: PMC10072234.

Methods:

This section comprises a lot of limitations

Participants:

The authors did not mention the basis for calculation of sample size.

Although authors described some features of the participants, they did not mention the exact inclusion and exclusion criteria of the sample. Neither the socioeconomic level nor the marital status of the participants mothers was elaborated; in spite of the known impact of these variables on the mother -infant interaction.

Study design:

The longitudinal study design allowed the authors to investigate the associations between mother-infant interaction characteristics at 9 months of age with maternal depression and infant temperament along the first year postpartum, and with child behavior at 3 years of age. It is important to consider the authors’ efforts to keep the sample intact over this time.

The authors mentioned that there were five points of assessment consisting of one pre- and four post-natal assessment points. However, they excluded the prenatal assessment point. I think it is important to include this assessment to avoid loss of the biological factors’ effect on the mother-infant interaction.

The authors must specify the time of the four post-natal assessment points in the section of methods. These points appeared for the first time in Table S1 in supporting information file.

The major limitation in methods was the single point assessment of the main outcome variable (mother-infant interaction). It was assessed once at the age of 9 months. This main variable must be evaluated as the average of several assessments at several points of time as the maternal depression and infant temperament. In addition, it was unsuitable to depend on the undergraduate students in evaluation of the mother-infant interaction. Instead, expert developmental and behavioral pediatricians had to do this evaluation.

The Strengths and Difficulties Questionnaire (SDQ) should not be used for this young age, many available questionnaires could be used to investigate the child’s behaviors, emotions and relationships at the age of 36 months. (e.g. Child Behavior Checklist (CBCL), Ages & Stages Questionnaire: Social-Emotional (ASQ: SE)

Confounding variables were not considered in this study, which might affect mother-infant relationship as health status of the infant or the mother, socioeconomic variables, social support, work status, and marital status.

Results:

Were well presented.

Table S6 in the supporting file: It described the associations between infant Falling Reactivity at each time point and Dyadic Affective Mutuality observed at 9 months. These associations were reported to be positive. Thus, authors have to revise for the negative sign in each correlation.

Discussion:

The finding of a lack of association between interaction behaviors and maternal depressive symptoms was not acceptable. This may be due to the dependence on single evaluation of interaction behavior as mentioned before; or because all participants had minimal depression symptoms that could not impact the interaction behavior. The other two findings are acceptable.

Reviewer #2: this study is very interesting for readers and caregivers, all the sections are very well described and data were collected methodologically correctly.

those aspects were very well descibed in the methods and result section

6. PLOS authors have the option to publish the peer review history of their article (what does this mean?). If published, this will include your full peer review and any attached files.

Reviewer #1: **Yes: **Ebtissam M. Salah El-Din

Reviewer #2: **Yes: **erich cosmi

---

## [Author Response · Author response to Decision Letter 0]

15 Mar 2024

PONE-D-23-29119

Mother-infant interaction characteristics associate with infant falling reactivity and child peer problems at pre-school age

PLOS ONE

Reviewers' comments:

Reviewer #1: Thank you for the opportunity to review the manuscript, “Mother-infant interaction characteristics associate with infant falling reactivity and child peer problems at pre-school age”. PONE-D-23-29119

The topic of the manuscript is interesting, the quality of the early interaction between an infant and his or her mother is an important factor that influences the child’s development, survival and growth. This type of research has been discussed repeatedly in previous literature; however, it is still needed to direct the plans for intervention procedures to alter mother–infant interaction with the ultimate goal of achieving positive developmental outcomes during the first years of life.

Response: We would like to thank this reviewer for acknowledging the importance of our work. 

The introduction

1. The introduction must be revised as there are some outdated references. It is prolonged and full of details of previous studies. It included description of the tools which have been used in the study. These details must be transferred to the methodology section.

These are examples of recent references that can help in revising the introduction

• Rocha, N. A. C. F., Dos Santos Silva, F. P., Dos Santos, M. M., & Dusing, S. C. (2020). Impact of mother-infant interaction on development during the first year of life: A systematic review. Journal of child health care: for professionals working with children in the hospital and community, 24(3), 365–385. https://doi.org/10.1177/1367493519864742

• Chung, FF., Wan, GH., Kuo, SC. et al. Mother–infant interaction quality and sense of parenting competence at six months postpartum for first-time mothers in Taiwan: a multiple time series design. BMC Pregnancy Childbirth 18, 365 (2018). https://doi.org/10.1186/s12884-018-1979-7

• Richter M, Fehringer K, Smith J, Pineda R. Parent-infant interaction in the NICU: Challenges in measurement. Early Hum Dev. 2022 Jul;170:105609. doi: 10.1016/j.earlhumdev.2022.105609. Epub 2022 Jun 14. PMID: 35752043; PMCID: PMC10072234.

Response: We are grateful to this reviewer for suggesting additional references and improvements to the Introduction section. We added the reference to the review by Rocha et al. (2020) which we found very relevant. However, we felt that Chung et al. (2018) and Richter et al.’s (2022) studies are too specific to special populations and therefore not directly relevant for our study – i.e., first-time mothers and high-risk dyads in NICU. As further recommended by this reviewer, the Introduction section has now been shortened in several places (pp.4-6) and details pertaining to methods originally included at the end of the Introduction section have been moved and integrated in the Methods section (pp.13-14).

Methods:

2. This section comprises a lot of limitations

Participants:

The authors did not mention the basis for calculation of sample size.

Response: Thank you for this important comment. When we started this longitudinal study, no formal sample size calculation was used due to the lack of previous longitudinal studies with similar measures to base the calculation on. However, in cross¬-sectional studies more generally, all the effects we were studying in terms of infants' attention and social abilities have been robustly found and replicated in samples much smaller (typically 10¬-20 infants) than the one we originally proposed (approx. 60). Since no previous study had investigated performance on the tasks longitudinally, we felt that a larger sample was needed to accommodate for participant drop¬out and the yet unknown magnitude of longitudinal stability during the first year of life.

The final sample size of our study was limited to some extent by the labour-intensive nature of carrying out longitudinal infant work (in terms of continuous participant contact, preparation for testing, testing with plenty of breaks to capture each infant's most alert and attentive state, and manual data analysis and coding of electrophysiological and behavioural data). 

We have added this sentence on p.8-9. “No formal sample size calculation was used due to the lack of previous longitudinal studies with similar measures to base the calculation on. There were also financial and time constraints. However, a sample size of 54 infants is larger than most infant studies involving extensive behavioural testing.”

3. Although authors described some features of the participants, they did not mention the exact inclusion and exclusion criteria of the sample. Neither the socioeconomic level nor the marital status of the participants mothers was elaborated; in spite of the known impact of these variables on the mother -infant interaction.

Response: We are sorry for this omission. At time of recruitment of our longitudinal study, the inclusion criteria were that the potential participant was female, pregnant, and had not yet entered the 37th week of pregnancy. There were no exclusion criteria. We have added this information to the Participants section on p.7. In the same section, we also explain that 2 infants were excluded from the recruited sample due to complications at birth. This was the sole exclusion criterion at that stage.

The reviewer is right that sociodemographic variables may play a critical role in mother-infant interactions, and these are summarised in the methods section on p.9. Each regression analysis was indeed run twice, once with and once without including sociodemographic variables. Since both analyses yielded comparable results, in the main text we only reported the analyses where the sociodemographic variables were excluded. We have added the analyses including the sociodemographic variables in the Supplementary Materials (Tables 8-11). This is now specified at the beginning of the Results section on p. 15-16. 

In addition to the regression analyses, we have now also checked the correlations between maternal depressive symptoms and the interaction behaviours considering the sociodemographic variables and observed a negative association between maternal depressive symptoms and MRIA (r=-.297, p=.047) and a positive association between maternal depressive symptoms and IN (r=.335, p=.024). Neither of these remained significant after controlling for multiple comparisons. These results have been added in the main manuscript on p. 18. Results are summarised in Table S9 in Supplementary Materials. We have now added a discussion of these findings on p.25-26.

Study design:

4. The longitudinal study design allowed the authors to investigate the associations between mother-infant interaction characteristics at 9 months of age with maternal depression and infant temperament along the first year postpartum, and with child behavior at 3 years of age. It is important to consider the authors’ efforts to keep the sample intact over this time.

The authors mentioned that there were five points of assessment consisting of one pre- and four post-natal assessment points. However, they excluded the prenatal assessment point. I think it is important to include this assessment to avoid loss of the biological factors’ effect on the mother-infant interaction.

Response: We would like to thank this reviewer for their thorough reading of our experimental procedure. We agree with the reviewer that collecting prenatal information is important. During the prenatal assessment in our study, we collected fetal heart rate and movements, but this data is still to be processed. We did not collect mothers’ depressive symptoms prenatally so unfortunately we cannot include this information in our manuscript. We have now included this information on p.8. We have also added a sentence in our limitations section on p.29: “Relatedly, collecting measures of prenatal maternal depressive symptoms as well as considering other (prenatal) biological factors will also be important in future studies.” 

5. The authors must specify the time of the four post-natal assessment points in the section of methods. These points appeared for the first time in Table S1 in supporting information file.

Response: We are sorry for this omission. The exact times of the four post-natal assessment points are now specified on p.8.

6. The major limitation in methods was the single point assessment of the main outcome variable (mother-infant interaction). It was assessed once at the age of 9 months. This main variable must be evaluated as the average of several assessments at several points of time as the maternal depression and infant temperament. In addition, it was unsuitable to depend on the undergraduate students in evaluation of the mother-infant interaction. Instead, expert developmental and behavioral pediatricians had to do this evaluation.

Response: We agree with this reviewer that it is more difficult to draw conclusions from a single observation than a series of repeated observations. However, the single behavioural assessment at the age of 9 months was originally designed as main outcome measure of our infant longitudinal study and was carried out with great attention and care. We are therefore confident that it is valid and provides relevant information about child development and the mother-child relationship. We are currently conducting research where mother-infant interaction is observed longitudinally across multiple time points, but this was unfortunately not possible in the present study due to resource constraints and the fact that the protocol included several other key developmental constructs of interest. Specifically, our study also includes comprehensive longitudinal measures and tasks of attentional and social development, in addition to the measures of maternal depressive symptoms, mother-infants interaction and infant temperament development. 

Regarding the coding of the mother-infant behaviours during interactions, we would like to assure this reviewer that the coders were fully and carefully trained by the first and second author following specific guidelines, and good interrater reliability was indeed achieved for most of the scales (p. 15). The research teams involved in carrying out this study have extended experience in working with undergraduate students and their guidance in behavioral coding use appropriate and rigorous training and data quality evaluation procedures. 

7. The Strengths and Difficulties Questionnaire (SDQ) should not be used for this young age, many available questionnaires could be used to investigate the child’s behaviors, emotions and relationships at the age of 36 months. (e.g. Child Behavior Checklist (CBCL), Ages & Stages Questionnaire: Social-Emotional (ASQ: SE)

Response: We acknowledge this reviewer’s view that there are other tools that can be used to investigate children’s behaviours at 3 years of age. However, while the Strengths and Difficulties Questionnaire (SDQ) was originally designed to assess behaviours, emotions and relationships in children and young people aged 4–17 years, it has previously been successfully used with parents (or educators) of 2–4-year-olds – as well as with youths aged 18 and over. In line with this, Goodman (2001) refers to the SDQ as a brief measure of prosocial behavior and psychopathology of 3–16-year-olds that can be completed by parents, teachers, or youths. There have also been previous studies which used the SDQ with younger children (e.g., Du et al., 2008).

We would furthermore like to draw this reviewer’s attention to a study by Goodman and Scott (1999) who directly compared the SDQ and the CBCL. They found that scores from the SDQ and CBCL were highly correlated and equally able to discriminate psychiatric cases from controls. They also report that the SDQ was significantly better than the CBCL at detecting inattention and hyperactivity, and at least as good at detecting internalizing and externalizing problems. Also, mothers of low-risk children were twice as likely to prefer the SDQ. Taken together, we are confident that the SDQ is a valid and appropriate measure for our study. 

References:

Du, Y., Kou, J., & Coghill, D. (2008). The validity, reliability and normative scores of the parent, teacher and self report versions of the Strengths and Difficulties Questionnaire in China. Child and adolescent psychiatry and mental health, 2(1), 1-15.

Goodman, R. (2001). Psychometric properties of the strengths and difficulties questionnaire. Journal of the American Academy of Child & Adolescent Psychiatry, 40(11), 1337-1345

Goodman, R., & Scott, S. (1999). Comparing the Strengths and Difficulties Questionnaire and the Child Behavior Checklist: is small beautiful?. Journal of abnormal child psychology, 27, 17-24.

8. Confounding variables were not considered in this study, which might affect mother-infant relationship as health status of the infant or the mother, socioeconomic variables, social support, work status, and marital status.

Response: As mentioned in our response to point 3 above, these variables were collected and considered in our preliminary analyses, and are now also added in the Supplementary Materials (Supp Tables 9-11). We have now specified this at the beginning of the Results section on p. 15-16, and added a discussion on the impact of SES in light of the associations with maternal depressive symptoms on p. 25-26.

Results:

Were well presented.

9. Table S6 in the supporting file: It described the associations between infant Falling Reactivity at each time point and Dyadic Affective Mutuality observed at 9 months. These associations were reported to be positive. Thus, authors have to revise for the negative sign in each correlation.

Response: We thank this reviewer for spotting this mistake. It has now been corrected. 

Discussion:

10. The finding of a lack of association between interaction behaviors and maternal depressive symptoms was not acceptable. This may be due to the dependence on single evaluation of interaction behavior as mentioned before; or because all participants had minimal depression symptoms that could not impact the interaction behavior. The other two findings are acceptable.

Response: We agree with this reviewer that a variety of factors might have precluded us from detecting a significant relation between maternal depressive symptoms and the child, mother, and mother-child interaction behaviours coded in this study. We now report and discuss in our manuscript that when we include SES factors in the analyses we find some correlations, though these do not survive multiple comparison correction. We addressed the point about a single evaluation point for the interaction in Point 6 above. Although our sample was not a clinical sample, the mean BDI-II score was 8.4 with a standard deviation of 6.7, clearly indicating that scores were not at floor and showed plenty of variation that maternal depressive symptoms could potentially associate with. We provide in-depth interpretations of both the significant and non-significant associations within the context of the existing literature in the Discussion on p. 24-25. 

Reviewer #2: this study is very interesting for readers and caregivers, all the sections are very well described and data were collected methodologically correctly.

those aspects were very well descibed in the methods and result section

Response: We thank this reviewer for their overall positive comments on our manuscript.

---

## [Decision Letter · Decision Letter 1]

2 Apr 2024

PONE-D-23-29119R1Mother-infant interaction characteristics associate with infant falling reactivity and child peer problems at pre-school agePLOS ONE

Dear Dr. Rigato,

Thank you for submitting your manuscript to PLOS ONE. After careful consideration, we feel that it has merit but does not fully meet PLOS ONE’s publication criteria as it currently stands. Therefore, we invite you to submit a revised version of the manuscript that addresses the points raised during the review process.

Following the revisions made, the paper has significantly improved. I encourage you to follow the suggestions of Reviewer 3, who has proposed some minor revisions. Although the reviewer has suggested small changes, I believe the issue that has been raised (the possibility of better understanding the changes between infancy and toddlerhood) is crucial, especially for interpreting the results obtained in this study.

We look forward to receiving your revised manuscript.

Kind regards,

Giulia Ballarotto

Academic Editor

PLOS ONE

Journal Requirements:

Reviewers' comments:

Reviewer's Responses to Questions

**Comments to the Author**

1. If the authors have adequately addressed your comments raised in a previous round of review and you feel that this manuscript is now acceptable for publication, you may indicate that here to bypass the “Comments to the Author” section, enter your conflict of interest statement in the “Confidential to Editor” section, and submit your "Accept" recommendation.

Reviewer #1: All comments have been addressed

Reviewer #2: All comments have been addressed

Reviewer #3: All comments have been addressed

2. Is the manuscript technically sound, and do the data support the conclusions?

Reviewer #1: Yes

Reviewer #2: Yes

Reviewer #3: Yes

3. Has the statistical analysis been performed appropriately and rigorously? 

Reviewer #1: Yes

Reviewer #2: Yes

Reviewer #3: Yes

4. Have the authors made all data underlying the findings in their manuscript fully available?

Reviewer #1: Yes

Reviewer #2: Yes

Reviewer #3: Yes

5. Is the manuscript presented in an intelligible fashion and written in standard English?

Reviewer #1: Yes

Reviewer #2: Yes

Reviewer #3: Yes

6. Review Comments to the Author

Reviewer #1: I’m pleased to review the updated version of this interested manuscript “Mother-infant interaction characteristics associate with infant falling reactivity and child peer problems at pre-school age”. PONE-D-23-29119R1

I would like to thank the authors as all previous comments have been addressed.

The new added paragraphs and explanation have enriched the manuscript.

I think this updated version is accepted and ready for publication

Reviewer #2: manuscript has been improved and suitable for publication. the authors addressed all the answers to revision and prepared the manuscript in order to fit fot those

Reviewer #3: Thank you for the opportunity to review the study entitled 'Mother-infant interaction characteristics associate with infant falling reactivity and child peer problems at pre-school age.' This study is very interesting and well-organized, addressing a topic that needs to emerge more in the literature. Although I have noticed that the authors have extensively revised the paper, significantly improving their manuscript, I believe it is important for the authors to further explore the developmental age they are addressing. Specifically, while the literature regarding mother-infant exchanges has been thoroughly explored, it would be important for the authors to also delve deeper into the changes that occur as the child grows. With the acquisition of greater autonomy and what M. Mahler has defined as a period of separation-individuation in the mother-child dyad, the child gains a greater sense of agency, which will continue to evolve during toddlerhood, the developmental phase addressed by the authors. It is important that this be better defined in the introduction and that the results be discussed based on the specific developmental context of the child and the dyad as a whole. Below, I highlight some literature that could be useful:

- Ballarotto (2023). Parental sensitivity to toddler's need for autonomy: An empirical study on mother-toddler and father-toddler interactions during feeding and play. Infant Behavior and Development

- Kemppinen (2006). The continuity of maternal sensitivity from infancy to toddler age. Journal of Reproductive and Infant Psychology

- Lecuyer-Maus (2000). Maternal sensitivity and responsiveness, limit-setting style, and relationship history in the transition to toddlerhood. Issues in Comprehensive Pediatric Nursing

- Ballarotto (2021). Mother-child interactions during feeding: A study on maternal sensitivity in dyads with underweight and normal weight toddlers. Appetite

- Andreadakis (2020). Toddler temperament, parent stress, and autonomy support. Journal of Child and Family Studies

Furthermore, in the study limitations, it is important to highlight the constraints of certain self-reports (such as the SDQ, particularly in situations of parental stress). You can refer to the study by:Trumello (2023) Mothers and fathers of pre-school children: a study on parenting stress and child’s emotional-behavioral difficulties

7. PLOS authors have the option to publish the peer review history of their article (what does this mean?). If published, this will include your full peer review and any attached files.

Reviewer #1: **Yes: **Ebtissam M. Salah El-Din

Reviewer #2: **Yes: **Erich Cosmi

Reviewer #3: No

---

## [Author Response · Author response to Decision Letter 1]

5 Apr 2024

PONE-D-23-29119R1

Mother-infant interaction characteristics associate with infant falling reactivity and child peer problems at pre-school age

PLOS ONE

Response to reviewers

Please find our answers in italics below each reviewer’s comment.

Reviewer #3: 

Thank you for the opportunity to review the study entitled 'Mother-infant interaction characteristics associate with infant falling reactivity and child peer problems at pre-school age.' This study is very interesting and well-organized, addressing a topic that needs to emerge more in the literature. Although I have noticed that the authors have extensively revised the paper, significantly improving their manuscript, I believe it is important for the authors to further explore the developmental age they are addressing. Specifically, while the literature regarding mother-infant exchanges has been thoroughly explored, it would be important for the authors to also delve deeper into the changes that occur as the child grows. With the acquisition of greater autonomy and what M. Mahler has defined as a period of separation-individuation in the mother-child dyad, the child gains a greater sense of agency, which will continue to evolve during toddlerhood, the developmental phase addressed by the authors. It is important that this be better defined in the introduction and that the results be discussed based on the specific developmental context of the child and the dyad as a whole. Below, I highlight some literature that could be useful:

- Ballarotto (2023). Parental sensitivity to toddler's need for autonomy: An empirical study on mother-toddler and father-toddler interactions during feeding and play. Infant Behavior and Development

- Kemppinen (2006). The continuity of maternal sensitivity from infancy to toddler age. Journal of Reproductive and Infant Psychology

- Lecuyer-Maus (2000). Maternal sensitivity and responsiveness, limit-setting style, and relationship history in the transition to toddlerhood. Issues in Comprehensive Pediatric Nursing

- Ballarotto (2021). Mother-child interactions during feeding: A study on maternal sensitivity in dyads with underweight and normal weight toddlers. Appetite

- Andreadakis (2020). Toddler temperament, parent stress, and autonomy support. Journal of Child and Family Studies

We are grateful to the Reviewer for their comments on our work and for pointing out the importance of highlighting the transition to toddlerhood with particular attention to the acquisition of greater autonomy. We have read the suggested references with interest and included those in the manuscript in the Introduction and in the Discussion when referring to our investigation of social and emotional outcomes at 3 years of age. Specifically, in the introduction we have added the references by Kemppinen et al. (2006) [37] and Lecuyer-Maus (2000) [43] and on p. 6 we have added the sentence: 

“More generally, and in accordance with a transactional perspective, during the transition to toddlerhood, mother-infant interactions lay the foundation for future developmental outcomes (e.g. [43]).”

In the Discussion we have expanded on the parents’ autonomy support findings citing Ballarotto et al. (2023) [81], Andreadakis et al. (2020) [87], and Joussemet et al. (2008) [80]. On p. 24 we have added: 

“Autonomy is considered a fundamental child need and parental support for autonomy is arguably one of the most important aspects of parenting for the development of child independence [80]. Requests for autonomy are often found among the most frequent cues in toddlers’ interactions with both their mother and father (e.g. [81]), highlighting the emerging need for autonomy in the transition from infancy to toddlerhood.” 

Later on in the paragraph we have added: 

“In addition, another recent study [87] reported differential patterns of associations between toddler temperament, parental stress and support for autonomy depending on the temperamental characteristics. Parent stress was found to fully account for the negative association between toddler negative affectivity and parent autonomy support, suggesting that toddler temperament affects parent wellbeing, which in turn is related to parenting practices.”

Furthermore, in the study limitations, it is important to highlight the constraints of certain self-reports (such as the SDQ, particularly in situations of parental stress). You can refer to the study by: Trumello (2023) Mothers and fathers of pre-school children: a study on parenting stress and child’s emotional-behavioral difficulties

We thank the reviewer for this additional point and reference. We have added this limitation on p. 25: 

“Collecting information from additional observers of the child (e.g., other family members, educators, etc.) would have allowed a more objective assessment of the infant temperament as well as the child’s social and emotional outcome at 3 years of age.”

---

## [Editor Report · Decision Letter 2]

10 Apr 2024

Mother-infant interaction characteristics associate with infant falling reactivity and child peer problems at pre-school age

PONE-D-23-29119R2

Dear Dr. Rigato,

We’re pleased to inform you that your manuscript has been judged scientifically suitable for publication and will be formally accepted for publication once it meets all outstanding technical requirements.

Kind regards,

Giulia Ballarotto

Academic Editor

PLOS ONE
---

## [Editor Report · Acceptance letter]

30 Apr 2024

PONE-D-23-29119R2 

PLOS ONE

Dear Dr. Rigato, 

I'm pleased to inform you that your manuscript has been deemed suitable for publication in PLOS ONE. Congratulations! Your manuscript is now being handed over to our production team.

Kind regards, 

on behalf of

Dr Giulia Ballarotto 

%CORR_ED_EDITOR_ROLE%

PLOS ONE